# Comparing Outcomes of Single-Incision Laparoscopic Herniorrhaphy in Newborns and Infants

**DOI:** 10.3390/diagnostics13030529

**Published:** 2023-02-01

**Authors:** Tsung-Jung Tsai, Ching-Min Lin, I Nok Cheang, Yao-Jen Hsu, Chin-Hun Wei, Tai-Wai Chin, Chin-Yen Wu, Wen-Yuan Chang, Yu-Wei Fu

**Affiliations:** 1Department of Surgery, Changhua Christian Hospital, Changhua 500, Taiwan; 2Division of Pediatric Surgery, Department of Surgery, Changhua Christian Hospital, Changhua 500, Taiwan; 3Division of Pediatric Surgery, Department of Surgery, Shuang Ho Hospital, New Taipei City 235, Taiwan; 4Department of Nursing, Changhua Christian Hospital, Changhua 500, Taiwan

**Keywords:** premature, newborns, infants, inguinal hernia, laparoscopy, outcomes

## Abstract

Background: As surgical techniques progress, laparoscopic herniorrhaphy is now performed more often in premature babies. The aim of this study was to analyze the outcomes of newborns and infants who underwent single-incision laparoscopic herniorrhaphy (SILH) at our center. Methods: We retrospectively reviewed patients younger than 12 months old who received SILH at our department from 2016 to 2020. SILH involved a 5 mm 30-degree scope and 3 mm instruments with a 3-0 Silk purse-string intracorporeal suture for closure of the internal ring. At the time of surgery, Group 1 newborns, whose corrected age was 2 months and below, were compared to the Group 2 infants, whose age was above 2 months. We assessed the patients’ characteristics, anesthesia, surgical data, and complications. Results: A total of 197 patients were included (114 newborns in Group 1 and 83 infants in Group 2). The mean age and body weight in Group 1 were 1.2 months and 3.8 kg, respectively, whereas in Group 2, they were 3.2 months and 6.7 kg, respectively. There were no significant differences in operative time (Group 1 = 34.1 min vs. Group 2 = 32.3 min, *p* = 0.26), anesthetic time (Group 1 = 80.0 min vs. Group 2 = 76.3 min, *p* = 0.07), length of hospitalization (Group 1 = 2.3 days vs. Group 2 = 2.4 days, *p* = 0.88), postoperative complications including omphalitis (Group 1 = 5.3% vs. Group 2 = 1.2%, *p* = 0.13), wound infection (Group 1 = 0.9% vs. Group 2 = 1.2%, *p* = 0.81), and hydrocele (Group 1 = 0.35% vs. Group 2 = 8.4%, *p* = 0.14). No recurrence, testicular ascent or atrophy, or mortality was observed in either group during the 2-year follow-up period. Conclusions: Single-incision laparoscopic herniorrhaphy is a safe and effective operation for inguinal hernia repair in infants, even those with prematurity, lower body weight at the time of surgery, or cardiac and/or pulmonary comorbidities. Comparable results revealed no significant differences in perioperative complications despite younger ages and lower body weights.

## 1. Introduction

In the 1990s, Schier F. first reported the laparoscopic approach for inguinal repair in children, and it has now become a commonly used technique [1]. There are various published benefits of laparoscopic herniorrhaphy (LH), including providing detection of and repairing metachronous inguinal hernias (MCIHs) without groin exploration, less manipulation of the spermatic cord, and better cosmetic results. There are also issues under debate regarding increased recurrence and complications as well as anesthetic risks associated with pneumoperitoneum in younger and smaller infants, especially those with anemia and with cardiac and respiratory comorbidities [2,3]. The timing of inguinal hernia repair in neonatal patients is a dilemma. The concerns relating to early repair include anesthetic risk and postoperative apnea or recurrence; for delayed repair, risk of incarceration is a major concern.

Several laparoscopic techniques were introduced and classified as internal ring suturing either extracorporeally or intracorporeally. The extracorporeal technique involves suturing with a needle inserted percutaneously, and the knot is tied outside the fascia. The intracorporeal technique involves suturing of the internal ring using various methods, including purse-string and Z-type. Percutaneous internal ring suturing is one of the extracorporeal techniques, and the operative time is less than that of intracorporeal techniques [4]. The learning curve is at least 30 cases [5]. A recent report on the learning curve of laparoscopic intra-corporeal inguinal hernia repair in children under the age of 16 years showed that it was 18 procedures [6]. Reports on the learning curve of LH in newborns are scarce in the literature. With improvements in technical skills, LH has evolved from a three-port to a single-incision approach. However, there are only a few studies addressing the outcome of LH in newborns [7,8]. After the SILH technique was proficiently conducted in older children in our center, we started to perform SILH in newborns in 2016. The aim of this study was to evaluate the safety and feasibility of SILH in newborns and infants and to compare the outcomes between different corrected ages at the time of surgery.

## 2. Materials and Methods

### 2.1. Patients

A retrospective cohort study of SILH was performed by two experienced pediatric surgeons at Changhua Christian Hospital. A total of 197 pediatric patients underwent hernia repair using the SILH technique from March 2016 to August 2020. The inclusion criteria were newborns and infants under the age of 12 months at the time of SILH, and if complications occurred, they underwent full follow-ups until resolution. The exclusion criteria included loss of follow-ups before the resolution of complication. Newborns whose corrected age was 2 months and below were assigned to Group 1, and infants whose corrected age was above 2 months were assigned to Group 2. This study was approved by the Institutional Review Board of our hospital on 2 February 2020, which waived the need for patient consent. Patient data including gender, prematurity, corrected age and body weight at the time of surgery, preoperative hemoglobin, side of inguinal hernia repair, anesthetic comorbidities (cardiac defects and lung disease), anesthetic time (measured as time from induction to extubation) and concurrent umbilical hernia were collected. Outcome measures included perioperative complications, omphalitis, surgical site infection, postoperative hydrocele, testicular ascent or atrophy, and recurrence of hernia. Our primary outcome was hernia recurrence; secondary outcomes included anesthesia time, perioperative complications, and cPPV.

### 2.2. Surgical Technique

All procedures were performed following the same standardized operative technique by two experienced pediatric surgeons specialized in laparoscopic hernia repair. Under general anesthesia with endotracheal intubation, the patient was placed in a supine position with their head angled slightly down. Two stay sutures were placed to hang up the umbilicus on either side (Figure 1a). A longitudinal incision was created at the center of the umbilicus, and the peritoneum was entered using an open technique. A subcutaneous tunnel about 2 cm in diameter was also created through the same incision at the same level on the right side of the umbilicus (Figure 1b). A 5 mm low-profile non-balloon umbilical port was used to induce pneumoperitoneum. Intra-abdominal pressure was kept at 6 to 8 mmHg CO_2_. A 5 mm 30-degree laparoscope was inserted to examine the patency of the processus vaginalis bilaterally (Figure 2a). If there was an internal ring defect, a long 3 mm trocar was stabbed through the subcutaneous tunnel into the abdomen under pneumoperitoneum with direct vision to avoid injury to the visceral organs. A 3–0 non-absorbable multifilamentous suture 17 mm 1/2 round-bodied needle (Mersilk, Ethicon, Blue Ash, OH, USA) was pierced directly through the abdominal wall lateral to the internal ring. The operator held the telescope in their left hand and the instrument in their right hand (Figure 1c). The defect was closed in a purse-string fashion starting laterally by skipping the vas deferens and the spermatic vessels under tension of the suture (Figure 2b–h). Before tying, the inguinal area was compressed gently to expel the gas in the hernia sac. After tying the knot intracorporeally, the suture was cut, and the needle was removed through the 3 mm trocar at the end of the procedure (Figure 2i). The port site was closed with a 2-0 Vicryl (Polyglactin 910, Ethicon Inc.) simple suture of the fascia and 4-0 Vicryl subcutaneously for the skin. Local anesthesia with bupivacaine was injected subcutaneously. Most infants over 50 gestational weeks of age were operated on in the day-surgery center and were discharged 1 h after surgery unless the parents wanted to admit their child for personal reasons. Patients under 50 gestational weeks of age were all admitted and observed after surgery. Postoperatively, the patients returned to the neonatal care nursery or infant care nursery for apnea monitoring. Simple analgesia (paracetamol) was used for pain relief. Patients were followed up with in a pediatric surgical outpatient clinic at 1 week, 1 and 12 months postoperatively to evaluate outcomes.

### 2.3. Statistical Analysis

Statistical evaluation was performed using SPSS for Windows, version 25 (SPSS Inc., Chicago, IL, USA, IBM group). A *p*-value less than 0.05 was assumed to represent statistical significance. The Kolmogorov–Smirnov test was used to test the normality of the data distribution. Patient demographics and outcomes were compared for the two groups. The data analyzed are expressed as mean and standard deviation. A two-sample *t*-test was used to compare the means of the continuous data. Percentages were calculated for categorical variables. The chi-square test or Fisher’s exact test were used to analyze categorical proportions.

## 3. Results

A summary of patient demographics and outcomes is listed in Table 1. SILH was performed in 197 infants, with 114 in Group 1 and 83 in Group 2. The male to female ratio in this study group was 3.9 (157 male patients and 40 female patients). There were 88 (77.1%) male newborns and 69 (83.1%) male infants. The mean age and body weight in Group 1 were 1.2 months and 3.8 kg, respectively, whereas in Group 2, they were 3.2 months and 6.7 kg, respectively. Age and body weight were significantly different between the two groups due to group selection (*p* < 0.01, two-sample *t*-test). There was no significant difference in laterality or umbilical hernia.

### 3.1. Underlying Diseases

Cardiac comorbidities included atrial and/or ventricular septal defects, and the main pulmonary comorbidity was chronic lung disease. In Group 1, there were 14 patients with significant comorbidities (14 cardiac, 0 pulmonary); in Group 2, there were 12 patients with significant comorbidities (10 cardiac, 2 pulmonary).

#### 3.1.1. Primary Outcome

##### Recurrence

No recurrence or MCIH was detected at a mean follow-up of 24 months (Table 2).

#### 3.1.2. Secondary Outcome

##### Anesthetic Time

There was no significant difference in mean anesthetic time between the two groups (80 (range 52 to 143) minutes in Group 1 compared with 76.3 (range 52 to 112) minutes in Group 2 (*p* = 0.07)).

### 3.2. Peri-Operative Complications

There were six cases of omphalitis in Group 1, which is defined as periumbilical erythema after postoperative day 7, and one in Group 2. One case of surgical site infection, defined as pus discharge, was found in both groups. There were four postoperative reactive hydrocele cases in Group 1 and seven in Group 2. Spontaneous resolution in 8 to 91 days was observed for all of these cases.

### 3.3. cPPV

The overall incidence of cPPV is 67.9%, which is higher than in previous studies. cPPV was found in 64 cases (56.2%) in Group 1 and 48 cases (34.1%) in Group 2; these values were significantly different (*p* < 0.01).

There were no intraoperative complications or need for conversion. All babies started feeding on the same day of surgery, and most of them were discharged from hospital the day after, except for four patients who had underlying cardiopulmonary diseases. There were no anesthetic complications and no mortalities in our cohort of patients.

## 4. Discussion

With the advancement of surgical instruments, more surgical techniques are proposed. In this study, we illustrated a new intracorporeal suture technique for newborn inguinal hernia along with a comparison to infants. No recurrence was noted during our two-year follow-up period. SILH in newborns was non-inferior to infants in terms of intraoperative and postoperative complications.

Pediatric inguinal hernias were traditionally repaired using an open approach with high ligation of the patent processus vaginalis at the level of the internal ring. In babies weighing <5 kg or less, even in experienced hands, traditional herniotomy can be a technically demanding procedure with an increased risk of postoperative complications (bleeding, wound infection, recurrence, persistent hydrocele, testicular ascent, and testicular atrophy) [1,9,10,11]. As surgical techniques progressed, minimally invasive laparoscopic approaches gained popularity, with comparable results to the open approach reported in the past two decades [1,12,13,14,15,16,17,18].

In the 1990s, El-Gohary and Schier F described the laparoscopic approach dealing with inguinal hernia in girls with inverted ligation or Z-sutures [1,19]. Monteput and Esposito reported laparoscopic hernia repair (LIHR) in boys with intracorporeal purse-string sutures [20]. In 2002, Schier F et al. reported a large retrospective review of multicentric experiences with encouraging results with LIHR [13]. In 2005, Harrison developed subcutaneously endoscopic assisted ligation (SEAL) as an alternative to early LIHR, which used multiple ports [21]. Oue T et al. introduced laparoscopic percutaneous extracorporeal closure (LPEC) as another acceptable alternative to the traditional approach [22]. In 2006, Patkowski et al. reported percutaneous internal ring suturing (PIRS), an innovative minimally invasive technique [17]. There were several modifications and refinements of the percutaneous technique that improved both the ease of the repair and patient outcomes. For examples, the double pass technique by Bruzoni et al. involves tenting the peritoneum off from the vas and vessels, thereby making the suture safer [23]; the Burnia technique by Novotny et al. involves cauterizing the PPV by inducing scarring at the internal ring in girls [24]; hydrodissection by Chen Y et al. involves injecting saline or bupivacaine to decrease injury to the cord and recurrence [25].

Since 2010, applications of LIHR in preterms and newborns have been reported by many authors. LIHR seems to be a safe, feasible, and effective procedure for preterms with a low recurrence rate [3,7,14,15,26]. Turial et al. reported a series of 147 babies weighing less than 5 kg with a reasonable recurrence rate (2%), no testicular atrophy, and 4% testicular ascent overall [14,15]. The same authors focused on preterm babies and reported a higher but still reasonable recurrence rate (3.6%), a relatively high testicular ascent rate (10%), and no testicular atrophy. Esposito et al. reported a series of neonates and preterms weighing less than 3 kg at surgery with a 4.4% recurrence rate, no testicular atrophy, and 15% testicular ascent [3]. Chan et al. reported a series of 79 premature neonates with a 1.3% recurrence rate [7]. Pastore et al. reported a series of 30 neonates with no recurrence [26].

LIHR is now being used worldwide with many refinements, and many centers have adopted the minimally invasive approach as the first choice for inguinal hernia repair in newborns. There were only a few studies considered comparing the specific subset group of newborns [27,28]. Therefore, we investigated our patients in this age group.

In our center, we have performed single-incision laparoscopic herniorrhaphy (SILH) to manage pediatric inguinal hernias since September 2015. Initially, we only performed it in girls at ages above 2 years. As more experience was gained, we applied this technique to boys, infants, and then premature infants. There are some points to be highlighted with our laparoscopic approach: (a) the single-incision wound allowed a 5 mm laparoscope and 3 mm instruments to work together well in a small space with good cosmesis (Figure 1a–c); (b) closure of the internal ring alone, leaving the distal sac without dissection, may decrease the operation time and not increase recurrence or hydroceles [16,29]; (c) using a skipped purse-string suture to close the internal ring decreased risks of injury to the vas deferens and vessels (Figure 2d) [17]; (d) using non-absorbable Mersilk sutures decreased the recurrence compared to the absorbable one [18,30]; (e) care was taken when tying the knot never to curl the vas deferens and spermatic vessels in, which can be assisted by pulling the testis down and tying slowly; (f) each tie was secured with three knots, enough to minimize the unfurling of the knots and hernia recurrence; (g) the suture ends were cut at least 5 mm away from the knots (measured by placing the scissor next to the suture to estimate this length before cutting); (h) adnexa sliding into the canal may retract into the abdomen by means of exterior stay sutures.

When performing laparoscopic hernia repair, cPPV can be found and closed simultaneously. The incidence of cPPV is age-dependent. Rowe et al. reported a 64% rate of cPPV identified at the time of inguinal hernia repair in infants younger than 2 months, and Saad et al. reported a rate of 44% in infants younger than 1 year [31,32]. There was a rate of 56.1% for cPPV in newborns and 34.1% in infants. The younger the age group, the higher the clinical incidence is, especially in male patients [33]. cPPV was age-specific and gender-specific, and many authors focused on the incidence and laterality of the younger patients. Liu reported that cPPV (46.95%) was more common in right-sided inguinal hernia in patients younger than 18 months [32]. Christine reported that preterm (50%) and term (42%) male patients with right side inguinal hernia had higher PPV [34]. We observed no such tendency, even though male patients were common in both groups (77.2% in Group 1 and 89.1% in Group 2). For newborns, cPPV was higher, and early surgery was preferred after a hernia was found.

Anesthesiologic issues were reported in the literature [3,35]. We observed a 5% rate of occurrence in term babies and 20% in preterms. The authors suspected that preexisting cardiopulmonary comorbidities could be risk factors. We analyzed perioperative indicators, such as preoperative hemoglobin, underlying diseases, intraoperative complications, and anesthetic and operation times in our series, which demonstrated no statistically significant differences. Many modern anesthesia techniques were applied to reduce the risks of general anesthesia [35,36]. Using a laryngeal mask for laparoscopic hernia repair in pediatric anesthesia resulted in a decrease in common complications and shortened anesthesia time [36]. A case series of caudal and intravenous anesthesia without intubation for LIHR was reported [37]. It could be a viable technique in the future.

The optimal timing of hernia repair in younger and smaller infants or preterm babies is debatable. The foreseeable complications of intra- and post-operation versus incarceration must be considered. Delayed repair may increase incarceration. Sulkowski reported that delayed repair in 667 patients displayed a 9.5% incarceration rate, whereas early repair had a higher reoperation rate within 1 year [38]. Some studies state otherwise. Osama A reported no difference in the risk of hernia incarceration and testicular atrophy in neonates between early or delayed repair [39]. Youn reported that early or delayed repair in preterm babies demonstrated no difference in preoperative incarceration and recurrence [40]. Our results revealed no postoperative respiratory complications, recurrences, or testicular atrophy, even among patients with cardiac or pulmonary disease. Hernia repair may be considered after anesthetic evaluation, even early repair.

There were some studies that reported long-term follow-up of laparoscopic inguinal hernia repair in children with modifications and refinements. Matias Bruzoni reported transfixation suture ligation in 166 patients with a 1.8% recurrence rate in a median 24-month follow-up, and the study concluded that this repair can be adopted by surgeons with basic laparoscopic skills and result in good outcomes with acceptable recurrence rates [41]. Denise I. Garcia reported on 10 years of experience with needle-assisted inguinal hernia repair in pediatric patients. A total of 1023 patients with a mean age of 2.5 years were included, with a 0.63% recurrence rate in preterm infants and four hydroceles requiring interventions with a mean follow-up of 5.9 years. The needle-assisted technique is safe and effective for preterm patients, with complication rates similar to those of other techniques [42]. Recently, a meta-analysis revealed the same recurrence rates following LH between preterm and full-term infants [43].

We did not encounter major perioperative complications, but we did encounter some minor ones. Omphalitis was observed in 5.3% of cases, more in newborns than in infants. This could be due to the short time since the separation of the umbilical cord, causing fragile skin and subcutaneous tissue with stich reactions (Figure 3). One recent study showed that direct transumbilical access in neonatal laparoscopic surgery is safe without increased risk for morbidity or mortality [44]. The rate of hydroceles was 3.5% in newborns and 8.4% in infants, and these spontaneously resolved during the follow-up. Closure of the internal ring in a skipped purse-string fashion without dissection of the distal sac is not associated with a higher risk of recurrence or hydrocele [17]. We did not observe any recurrence, testicular ascent, atrophy, or mortality. Regarding comorbidities, pneumonia and urinary tract infection occurred. One of patients with pneumonia had bronchopulmonary dysplasia; two had underlying cardiac conditions. Due to lower levels of comorbidity, no relation with other factors was found, but it may be common in patients with underlying diseases.

Finally, it was reported that open inguinal hernia repair is one of the most common risks for male infertility [45]. We should investigate the effects of laparoscopic IH repair on the male reproductive system by designing a long-term prospective study.

There are some limitations to this study. Retrospective selection bias cannot be avoided. All operations were performed by two laparoscopic surgeons, regardless of age and sex. The other limitation was a small number of patients with a short follow-up time. Because most recurrences and morbidities due to the operations usually present within the first postoperative year, we believe that the data may be reliable in terms of recurrences and postoperative complications.

The SILH technique is a safe, feasible, and effective operation for minimally invasive inguinal hernia repair in infants, including those who are premature, have a lower body weight at the time of surgery, and have cardiac and/or pulmonary comorbidities. A prospective, randomized controlled trial in infants comparing outcomes following laparoscopic versus open inguinal hernia repair will be needed in the future.

## Figures and Tables

**Figure 1 diagnostics-13-00529-f001:**
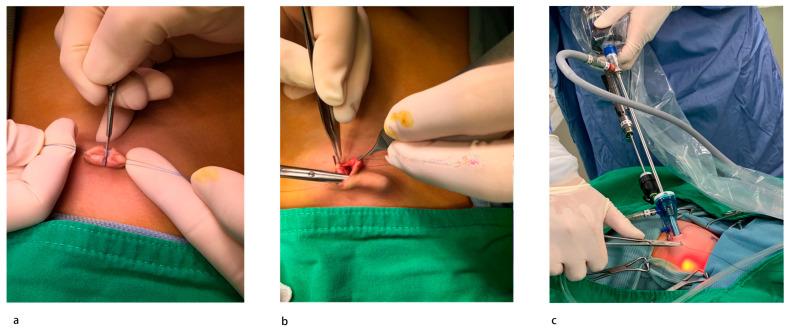
Single-incision laparoscopic herniorrhaphy setup: (**a**) make a midline incision on the umbilicus with stay sutures bilaterally; (**b**) create a subcutaneous tunnel through the same incision; (**c**) 5 mm trocar into the central wound, a 3 mm trocar in the tunnel.

**Figure 2 diagnostics-13-00529-f002:**
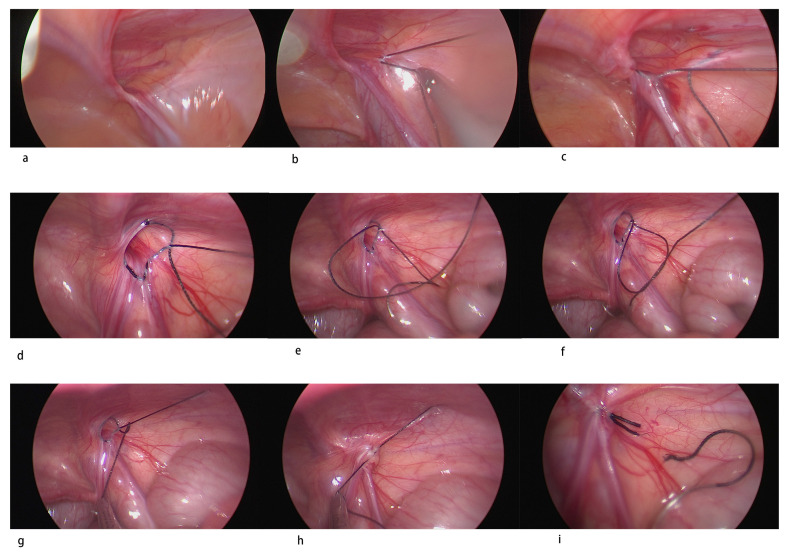
Step by step single-incision laparoscopic herniorrhaphy: (**a**) examine bilaterally; (**b**,**c**) skip the vas deferens and the spermatic vessels under tension; (**d**) complete intracorporeal skipped purse string suture; (**e**) pull the exterior string inside and make a cross; (**f**) put the needle into the circle and make a half stitch; (**g**,**h**) tie the knot cautiously so as not to curl the cord in; (**i**) cut the suture.

**Figure 3 diagnostics-13-00529-f003:**
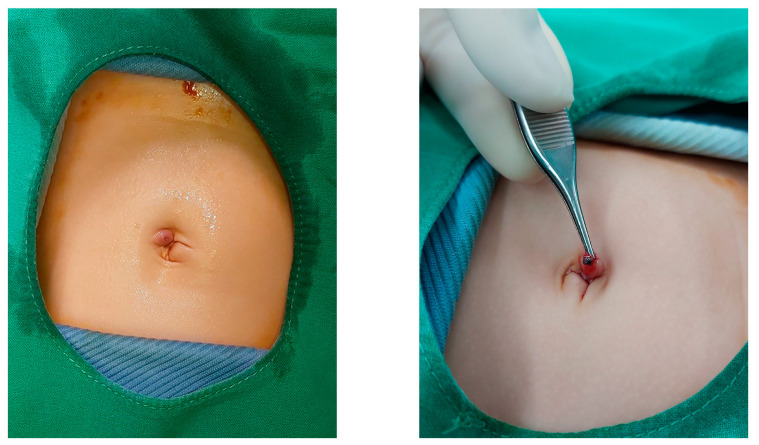
Stich reactions causing omphalitis.

**Table 1 diagnostics-13-00529-t001:** Patient group demographics.

Characteristics	Group 1 (Age ≤ 2 mo)				Group 2 (Age > 2 mo)				*p*-Value
No. of patients (*n*)	114				83				
Age (weeks)	4.9 ± 1.8				23.1 ± 15.9				<0.01 *
Gender (male)	88 (77.2%)				69 (89.1%)				0.31
Weight (kg)	3.8 ± 0.8				6.7 ± 2.1				<0.01 *
Laterality	Preoperative		Postoperative		Preoperative		Postoperative		
	Right:	51	Right:	14	Right:	48	Right:	19	
	Left:	36	Left:	9	Left:	27	Left:	8	
	Bilateral:	27	Bilateral:	91	Bilateral:	8	Bilateral:	54	
Umbilical hernia (*n*)	13 (11.4%)				12 (14.5%)				0.53
Pre-OP Hb (g/dL)	10.6 ± 1.7				11.4 ± 1.3				0.1
Underlying disease (*n*)									
Cardiac	14 (12.3%)				10 (12.0%)				0.1
Pulmonary	0				2 (2.4%)				

* Significant differences. cPPV, contralateral patent processus vaginalis; kg, kilogram; Hb, hemoglobin; LOS, length of stay.

**Table 2 diagnostics-13-00529-t002:** Intraoperative and postoperative outcomes.

Characteristics	Group 1 (Age ≤ 2 mo)	Group 2 (Age > 2 mo)	*p*-Value
Intraoperative complication	None	None	
Operative time (min)	34.1 ± 10.8	32.3 ± 11.0	0.26
Unilateral	28.8 ± 9.4	25.9 ± 8.5	
Bilateral	35.4 ± 10.8	35.4 ± 10.7	
Anesthetic time (min)	80.0 ± 14.5	76.3 ± 12.9	0.07
Unilateral	78.1 ± 16.5	72.2 ± 12.6	
Bilateral	80.4 ± 14.0	78.2 ± 12.7	
cPPV (*n*)	64 (56.1%)	48 (34.1%)	<0.01 *
cPPV-L	27(23.7%)	19(23.2%)	0.78
cPPV-R	37(32.5%)	29(35.4%)	0.21
Conversion	0	0	
Comorbidity (*n*)			
Pulmonary	2 (1.8%)	1 (1.2%)	
UTI	1 (0.7%)	0	
LOS (days)	2.3 ± 1.6	2.4 ± 9.3	0.88
Postoperative complication			
Omphalitis	6 (5.3%)	1 (1.2%)	0.13
Surgical site infection	1 (0.9%)	1 (1.2%)	0.81
Transient hydrocele	4 (3.5%)	7 (8.4%)	0.14
Resolution day	1.2 ± 9.0	13.5 ± 63.3	0.04
Recurrence (*n*)	0	0	
Testicular acent/atrophy	0	0	

* Significant differences. cPPV, contralateral patent processus vaginalis; cPPV-R, cPPV of right-sided inguinal hernia; cPPV-L, cPPV of left-sided inguinal hernia; LOS, length of stay.

## Data Availability

Not applicable.

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
