# Peer review of "Comparing Outcomes of Single-Incision Laparoscopic Herniorrhaphy in Newborns and Infants"

_diagnostics, 2023, doi:10.3390/diagnostics13030529_

Round 1
Reviewer 1 Report
The authors evaluated the treatment and outcome of neonates and infants who underwent single-incision laparoscopic herniorrhaphy at their center. They concluded that laparoscopic single-incision herniorrhaphy is a safe and effective operation for repairing inguinal hernias in infants, even if they were born prematurely, had lower body weight at the time of surgery, or had cardiac and/or pulmonary comorbidities.
I read the study with great interest. The study is interesting but needs considerable revision. My concerns are as follows
1. Abstract - The authors should present the results of the comparison variables along with the p - values. They made only general statements: ''There were no significant differences in operative time, anesthesia time, length of stay, postoperative complications including omphalitis, wound infection, and hydrocele...''. Please add the results and p - values for each variable measured.
2. The introduction is very short, the authors should add a few more lines related to the available laparoscopic techniques. For example they did not even mention percutaneous internal ring suturing – PIRS (reference: Percutaneous internal ring suturing for the minimal invasive treatment of pediatric inguinal hernia: A 5-year single surgeon experience. Surg Laparosc Endosc Percutan Tech. 2021 Apr 15;31(2):150-154. doi: 10.1097/SLE.0000000000000878.). Also they should add few lines regarding learning curves in laparoscopic hernia surgery (reference: Learning curve for laparoscopic repair of pediatric inguinal hernia using percutaneous internal ring suturing. Children (Basel). 2021 Apr 11;8(4):294. doi: 10.3390/children8040294.)
3. The authors presented outcome measures. Please separate which was the primary outcome and all others should be listed as secondary outcomes of the study.
4. Please provide clear inclusion/exclusion criteria for the study.
5. The Institutional Review Board statement and the reference and date of approval should be mentioned in the methodology.
6. Please indicate which statistical test was used to test normality of data distribution.
7. Results - Preoperative Hemoglobin Levels / Underlying Diseases - This is presented in Table 1 - There is no need to repeat the data from the table in the text of the results section. Please revise.
8. Discussion section needs to be rewritten/structured. Write in four consecutive paragraphs (without headings): (i) summary (not data) of the findings of this study; (ii) logical and coherent comparison with existing literature focusing on the main objective(s); (iii) limitations of the study; and (iv) Implications for practice/policy/research with a concluding statement.
9. Recently published meta-analysis on similar topic should be discussed and listed in reference list: Pogorelić Z, Anand S, Križanac Z, Singh A. Comparison of recurrence and complication rates following laparoscopic inguinal hernia repair among preterm versus full-term newborns: A systematic review and meta-analysis. Children (Basel). 2021;8(10):853. doi: 10.3390/children8100853.
10. The article would benefit from professional English proofreading.
Author Response
We thank the reviewer for the time and effort invested into reviewing our manuscript. Here are our response to your comments and feedback:

Reviewer 2 Report
Hi dear editor,
I would like to recommend a minor reversion of article candidate titled "Comparing Outcomes of Single-Incision Laparoscopic Herniorrhaphy in Newborns and Infants" before acceptance.
The novelty of this paper is significant and interested to the readers in the related field; the experiment is properly designed and complemented; the results are solid and the conclusion are clear. The surgical method and refinement were highlighted in this paper and the conclusion can promote current clinical performance.
However I would suggest a few modifications that will be the concerning to the readers.
1. The advantage and necessity of early surgery are the most vital foundation of this research, and were not fully illustrated in the paper. Please add it in the introduction and discussion part.
2. Please add the stratified analysis of several impact factors such as gender and the comorbidities; and consequently illustrate their impact on the surgical performance and results.
3. The overall cPPV were significant higher in the lower group age. Please balance this point with the advantage of surgery in early age.
In general, I would recommend a minor reversion before acceptance of this paper.
Author Response

(The authors gave the same response as above.)

Round 2
Reviewer 1 Report
The authors have improved the manuscript, but some of my objections have not been adequately addressed:
1. The authors stated in their revised introduction that the learning curve for percutaneous internal ring suturing is at least 30 cases, but they did not provide a reference to support this statement. Please revise! REFERENCE: Learning curve for laparoscopic repair of pediatric inguinal hernia using percutaneous internal ring suturing. Children (Basel). 2021;8(4):294. doi: 10.3390/children8040294.)
2. In response to my question about the outcomes of the study, the authors stated in their letter that their primary outcome is hernia recurrence; secondary outcomes include anesthesia time, perioperative complications, and cPPV, but they did not include this statement in their manuscript. This should be clearly stated in the methodology. Please revise.
3. My objection regarding the discussion was not adequately addressed. The authors responded that the paper was proofread in English, but the objection was about the structure of the discussion. The "discussion" section needs to be rewritten/structured. Write in four consecutive paragraphs (without headings): (i) summary (not data) of the findings of this study; (ii) logical and coherent comparison with existing literature focusing on the main objective(s); (iii) limitations of the study; and (iv) Implications for practice/policy/research with a concluding statement.
Author Response
We thank the reviewer for the time and effort invested into reviewing our manuscript. Here are our response to your comments and feedback